# A Hybrid Deep Learning-Based Approach for Brain Tumor Classification

Asaf Raza [1], Huma Ayub [1,*], Javed Ali Khan [2], Ijaz Ahmad [3,*], Ahmed S. Salama [4], Yousef Ibrahim Daradkeh [5], Danish Javeed [6], Ateeq Ur Rehman [7] and Habib Hamam [8,9,10,11]

[1] Department of Software Engineering, University of Engineering and Technology, Taxila 44000, Pakistan; asafraza94@gmail.com
[2] Department of Software Engineering, University of Science and Technology Bannu, Bannu 28100, Pakistan; engr_javed501@yahoo.com
[3] Shenzhen College of Advanced Technology, University of Chinese Academy of Sciences (UCAS), Shenzhen 518055, China
[4] Electrical Engineering Department, Faculty of Engineering & Technology, Future University in Egypt, New Cairo 11845, Egypt; a.salama@fue.edu.eg
[5] Department of Computer Engineering and Networks, College of Engineering, Prince Sattam Bin Abdulaziz University, Wadi Addawasir 11991, Saudi Arabia; y.daradkeh@psau.edu.sa
[6] Software College, Northeastern University, Shenyang 110169, China; 2027016@stu.neu.edu.cn
[7] Department of Electrical Engineering, Government College University, Lahore 54000, Pakistan; ateqrehman@gmail.com
[8] Faculty of Engineering, Uni de Moncton, Moncton, NB E1A3E9, Canada; habib.hamam@umoncton.ca
[9] International Institute of Technology and Management, Libreville BP1989, Gabon
[10] Spectrum of Knowledge Production & Skills Development, Sfax 3027, Tunisia
[11] Department of Electrical and Electronic Engineering Science, School of Electrical Engineering, University of Johannesburg, Johannesburg 2006, South Africa
* Correspondence: huma.ayub@uettaxila.edu.cn (H.A.); ijaz@siat.ac.cn (I.A.)

**Abstract:** Brain tumors (BTs) are spreading very rapidly across the world. Every year, thousands of people die due to deadly brain tumors. Therefore, accurate detection and classification are essential in the treatment of brain tumors. Numerous research techniques have been introduced for BT detection as well as classification based on traditional machine learning (ML) and deep learning (DL). The traditional ML classifiers require hand-crafted features, which is very time-consuming. On the contrary, DL is very robust in feature extraction and has recently been widely used for classification and detection purposes. Therefore, in this work, we propose a hybrid deep learning model called DeepTumorNet for three types of brain tumors (BTs)—glioma, meningioma, and pituitary tumor classification—by adopting a basic convolutional neural network (CNN) architecture. The GoogLeNet architecture of the CNN model was used as a base. While developing the hybrid DeepTumorNet approach, the last 5 layers of GoogLeNet were removed, and 15 new layers were added instead of these 5 layers. Furthermore, we also utilized a leaky ReLU activation function in the feature map to increase the expressiveness of the model. The proposed model was tested on a publicly available research dataset for evaluation purposes, and it obtained 99.67% accuracy, 99.6% precision, 100% recall, and a 99.66% F1-score. The proposed methodology obtained the highest accuracy compared with the state-of-the-art classification results obtained with Alex net, Resnet50, darknet53, Shufflenet, GoogLeNet, SqueezeNet, ResNet101, Exception Net, and MobileNetv2. The proposed model showed its superiority over the existing models for BT classification from the MRI images.

**Keywords:** deep learning; brain tumor; MRI; transfer learning; convolutional neural network

## 1. Introduction

The human brain is a command center and an essential organ of the human nervous system responsible for accomplishing daily life activities. The brain collects stimuli or signals from the body's sensory organs, handles processing, and directs the ultimate

decisions and output information to the muscles. BTs is one of the most severe situations related to the human brain, where a group of abnormal brain cells grows in an undisciplined manner [1]. BTs can be divided into two main types: primary and secondary metastatic. The primary brain tumors (BTs) are generally non-cancerous and originate from human brain cells. In contrast, secondary metastatic tumors spread to the brain with blood flow from other body parts.

Furthermore, the World Health Organization (WHO) classified BTs into four categories (Grade I–IV) depending on their malignancy or benignity. The standard approaches for detecting and analyzing BTs are magnetic resonance imaging (MRI) and computer tomography (CT) [2]. Grade III and Grade IV malignant BTs are fast-growing; they spread to other body parts and affect healthy cells. Thus, early BT detection and classification helps doctors plan proper treatment based on MRI and other images [3]. Glioma, pituitary and meningioma are the three main types of primary brain tumors. Pituitary BTs are generally benign and grow in the pituitary glands, the base layer of the brain that produces some essential hormones in the body [4]. Gliomas develop from the glial cells of the brain [5]. Meningioma tumors generally grow on the protective membrane of the brain and spinal cord [6]. The separation of normal brain tissue from abnormal tissues is critical in BT detection. Due to size, shape, and location variations, BT detection becomes more energizing and is still an open problem. The concepts of medical image processing are used in BT analysis (i.e., classification, segmentation, and detection) [7]. BT classification is a necessary procedure to identify the tumor type at an early stage, if there are any. Many modernistic, computer-aided diagnosis systems are presented in biomedical image processing to help radiologists guide patience and better classify a BT [8]. A BT is a hazardous disease, and it causes shorter life when there are high-grade tumors. To be precise, the diagnosis of a BT plays a vital role in treatment and is helpful for the patient's life [9]. Due to high variance, low contrast in nasopharyngeal carcinoma (NPC), and disrupted edges in magnetic resonance images (MRIs). Accurate tumor segmentation is critical in the guidance of a radiologist [10] to identify tumors better. There are numerous deep learning architectures proposed in the literature for BT segmentation, such as DensNet [11], ResNet [12], and InceptionNet [13].

In the literature, ML and DL are the two main techniques implemented for BT detection [14,15]. Various studies have been proposed that employed machine learning methods, such as support vector machines (SVM) [16,17], k-nearest neighbor (KNN) [18], principal component analysis (PCA) [19], decision trees, and artificial neural networks (ANNs) [20,21]. However, these methods work on hand-crafted features, while the mean features need to be extracted for the training process. Therefore, the detection and classification accuracy depend on the quality of the features. Machine learning classifiers are time-consuming and require large memory for large datasets [22]. Additionally, CNN layers are widely used for image and speech feature extraction [23]. Artificial neural networks are also used for the extraction of different features, as each neuron is connected to another neuron [24]. However, in deep learning, the last layers are fully connected and perform well in medical imaging. For example, the CNN is the most common DL model mostly used for image classification [25].

This inspired us to propose a DL-based approach to enhance existing algorithms' accuracy and performance in detecting various types of BTs and evaluate the approach on a publicly available dataset. For this purpose, we propose a hybrid DeepTumorNet model that identifies and classifies BTs under the following three main types: meningiomas, pituitary, and gliomas. The proposed method adopts the mechanism of deep learning for feature extraction and a Softmax classification layer for variety. The proposed model recorded the highest ever classification accuracy on the (CE-MRI) dataset, which is publicly available on figshare, compared with the traditional method (i.e., Google net [26], Alex net [27–29], Resnet50 [12], Squeezenet [28], DensNet [11], darknet53 [29], Mobilenetv2 [30], Resnet101 [31], and Shufflenet [32]). Furthermore, with the proposed research study, we

aim to answer the following research question: How efficiently and correctly does the deep learning algorithm identify and classify BTs into different BT diseases?

Our main contribution to this research includes the following.

We propose a hybrid deep learning model for three types of BT classification (pituitary, meningiomas, and gliomas), where the proposed DL classification approach outperforms the existing state-of-the-art approaches by showing the highest accuracy on the CE-MRI dataset, and extensive experiments are performed with nine other pretrained deep learning models using transfer learning techniques. We also compare the results of the proposed model with the previously proposed methods.

This research article is organized as follows. Section 2 presents the related works. Section 3 details the proposed methodology. Section 4 describes the results and discussion. Section 5 gives the conclusion and future work as a result of this study.

## 2. Related Works

Recent human works on computer-aided medical diagnosis provide improved performance. For instance, Gupta et al. [33] integrated canny detection with an ANN for BTs identification. They adjusted the sizes of previously treated images by RGB2grey conversion and implemented a clever edge detection method to remove the qualities from BTs pictures. In the end, an ANN was implemented for BTs identification. Mehrotra et al. [34] proposed a DL network for two classes (malignant and benign) of dataset classification. They utilized five different pre-trained deep learning models (AlexNet, GoogLeNet, ResNet101, ResNet50, and SqueezeNet). AlexNet accomplished the most acceptable accuracy of 99.04%. Moreover, it was observed that the model depended on the chosen optimizer.

Ramzan et al. [35] proposed a DL model for BTs detection using a fusion of hand-crafted and deep features. The authors segmented BTs by applying a grab cut algorithm and some morphological operations. Moreover, deep learning (VVG19) and handmade components extracted for the tumor segments were fused through a serial-based method. These concatenated features were supplied to multiple classifiers. Raja et al. [36] enhanced the MR images using the nonsubsampled contourlet transform (NSCT) and NSCT fused separately. Low- and high-frequency sub-bands were connected independently.

Furthermore, the inverse NSCT was applied to obtain enhanced images and then extract the textured features from the enhanced images. The adaptive neuro-fuzzy inference system (ANFIS) method was used to classify these features into normal and glioma BT images. The classified glioma images were segmented using a morphological operation.

Sultan et al. [37] proposed a DL model based on a CNN using two publicly available datasets containing 3064 (meningioma, glioma, and pituitary tumors) and 516 (Grade II, Grade III, and Grade IV) images. The proposed model achieved a highest accuracy of 96.13% and 98.7%, respectively. Kumar et al. [38] introduced a hybrid approach for BT detection and classification. The proposed strategy extracted the features from MR images by applying a discrete wavelet transform and using PCA for feature reduction. Using a kernel SVM trained with the reduce feature, the proposed hybrid performed better than traditional DL methods by increasing the accuracy and decreeing the root mean square error. Pitchai et al. [39] integrated an artificial neural network (ANN) along with a fuzzy k-means algorithm for BT segregation and detection. The GLCM-extracted features are given to the ANN for a variety of standard and abnormal MRI images.

Furthermore, the fuzzy k-means algorithm segregates tumors from abnormal MRIs. Amin et al. [40] proposed an automatic detection and classification method for three publicly available BT datasets. Their proposed strategy comprises three steps. They segment the region of interest using different pre-processing techniques and morphological operations. Moreover, each candidate's lesion intensity, texture, and shape features are extracted and classified with an SVM. The proposed technique is more robust than other existing methods and returned 97.1% accuracy. Özyurt et al. [41] proposed a BT detection method. First, they segmented the MRI tumor images using the NS-EMFSE method. They extracted features from the segmented images using AlexNet and then the ML model, and KNN and a SVM

were utilized to detect and classify the BT images as benign or malignant using the SVM, achieving 95.62% accuracy. Kaplan et al. [42] modified the local binary patterns (LBP) feature for BT detection and classification. They extracted the nLBP and αLBP features and performed classification with different ML classifiers such as KNN, random forest (RF), ANN, A1DE, and linear discriminant analysis (LDA). Their proposed method achieved the highest accuracy of 95.56%, and nLBP d = 1 feature extraction with the KNN model, where d is the distance between two neighbors' pixels.

Çinar et al. [43] proposed a hybrid CNN architecture for BT detection by modifying a deep learning model. They removed the last five layers of ResnNet50 and added eight new layers. Their proposed hybrid ResNet50 model obtained 97.2% accuracy, while the single ResNet50 model obtained 92.53% accuracy. Mohsen et al. [44] introduced a DNN for BT detection and classification, where the MR images are segmented into normal and abnormal by using the fuzzy c-means clustering technique. The features are extracted from segmented images using DWT.

Moreover, these features are reduced by using PCA. Finally, classification is performed by a deep neural network. Like a CNN, the proposed methodology achieved the highest accuracy of 98.4%. Lather et al. [45] investigated different BT segmentation and detection techniques. Their study surveyed six segmentation techniques and presented a detailed review of previous researchers' work on segmentation and detection. Abd-Ellah et al. [46] suggested two-phase models for BT detection and localization. To detect brain tumors, the proposed system in the first phase classifies MR images as normal or abnormal by using a CNN for features extraction and the ECOC-SVM approach for classification. In the second phase, the tumor is localized in strange images by designing a new five-layer region based on R-CNN. The proposed two-phase multi-model achieved 99.5% detection accuracy. Marghalani et al. [47] proposed an automatic classification method for BT and Alzheimer's disease. They extracted a bag of features (SURF and SIFT) and used a SVM as a classifier.

The proposed automatic classification method classified the MRI data set into three classes—MRIs of brain tumors, MRIs of Alzheimer's disease, and MRIs of normal brains—with an average accuracy of 97%, which was greater than that of the SURF-based features. Swati et al. [48] used block-wise fine-tuned CNN models for BT detection and classification. The fine-tuned VGG-19 for the block-wise fine-tuning technique achieved 94.84% classification accuracy in less training time than the hand-crafted features. Kumar et al. [49] introduced a new optimized DL mechanism for BT detection and classification named Dolphin-SCA based on a deep CNN. For segmentation, the researcher used a fuzzy deformable fusion model with a dolphin echolocation-based sine cosine algorithm (Dolphin-SCA). The extracted features were used in a deep neural network with Dolphin-SCA based on the power LDP and statistical extracted features. The proposed technique achieved 96.3% classification accuracy.

Deepak et al. [50] used pre-trained GoogLeNet for feature extraction and proven classifier models for BT detection and classification. The proposed approach recorded 98% accuracy compared with the state-of-the-art methods. Raja et al. [51] presented a hybrid model for BT detection and classification that involves different backgrounds (i.e., pre-processing using a non-local express filter and segmentation using the Bayesian fuzzy method). Afterward, various image features were captured and extracted using theoretic measures, scattering transform, and the wavelet packet Tsallis entropy method. Finally, classification was carried out using a hybrid approach based on deep autoencoder with a Softmax regression and obtained 98.5% accuracy. Rammurthy et al. [52] proposed a new BT detection technique based on DL, namely Whale Harris Hawks optimization, by combining the whale optimization algorithm (WOA) with the Harris hawks optimization (HHO) algorithm. At first, the tumors in the images are segmented using cellular automata, and different features like the size, variance, mean, and kurtosis are extracted, while the elements are classified for better BTs detection with the proposed Whale Harris Hawks optimization (WHHO). The proposed method reached its highest accuracy at 81.6%. Bahadure et al. [53] used a skull skipping algorithm to eliminate the non-brain parts from

MR images to detect BTs based on Berkeley wavelet transformation (BWT) and segmented features (shape, texture, color, and contrast) with an SVM classifier. The experimental results obtained 96.51% accuracy. Waghmare et al. [54] implemented different CNN architectures for BTs detection and classification. Fine-tuned VGG-16 increased the classification accuracy of the augmented data set and reached the most acceptable accuracy at 95.71%.

## 3. Materials and Methods

This section highlights and elaborates the proposed research methodology for fine-grained BTs classification. The proposed method is mainly described in two steps. First, we elaborated on the research dataset utilized for fine-grained BTs classification. Secondly, we thoroughly elaborated on the proposed deep learning-based approach and its architecture to detect and classify brain MRI images into meningiomas, gliomas, and pituitary tumors.

### 3.1. Dataset and Preprocessing

This research study utilized a publicly available CE-MRI dataset [55]. The MRI images in the dataset were collected over 5 years (2005–2010) from 233 different patients having BTs at Nanfang Hospital Guangzhou China and General Hospital Tianjin Medical University in China. The dataset comprises 3062 MRI images of 3 distinct types of BT in 233 patients, including gliomas (1426), meningiomas (708), and pituitary tumors (930) in 3 different views. The details of the research dataset are explained in Figure 1 and Table 1. The dataset images are in 2D volumes and have a $512 \times 512$ resolution with a $0.49 \times 0.49$ mm$^2$ pixel size. Additionally, the tumor region in the MRI was bordered manually by three experienced radiologists [56]. The dataset images available on figshare are in .mat format and have a $512 \times 512$ resolution. The proposed model was designed with an input layer size of $224 \times 224$. Therefore, the dataset was pre-processed to make it pursuable for the proposed approach, as shown in Figure 2. Initially, the MRI images were normalized (i.e., mat to .jpg format) and then resized by using the resize function in Matlab [57] according to the image input sizes of our proposed deep learning model and other pretrained models. Therefore, the MRI images were resized to $224 \times 224$ pixels, and the DarkNet19 images were resized to $256 \times 256$ pixels. Furthermore, the dataset images were split into 70% for training and 30% for testing. We used all the 3064 brain tumor images for the experiments, where around 2146 images (495 meningiomas, 652 pituitary tumors, and 999 gliomas) were used for training. The remaining 918 images (213 meningiomas, 278 pituitary tumors, and 428 gliomas) for testing.

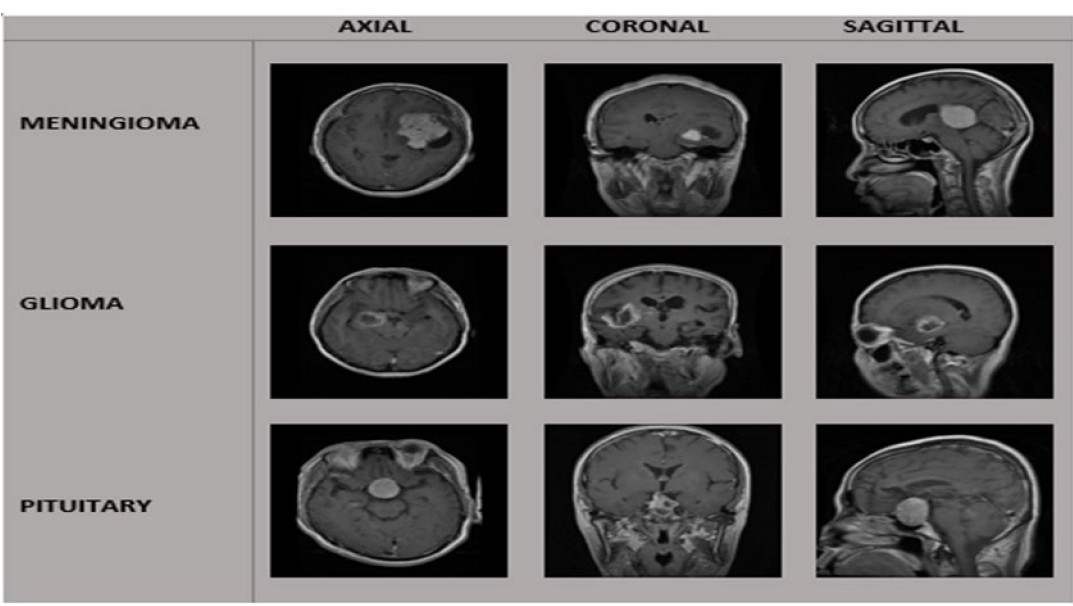

**Figure 1.** Three different tumors (meningioma, glioma, and pituitary tumor) in three different views.

**Table 1.** Explanation of CE-MRI dataset.

| Tumor Class | Patients | Images | View of MRI. | No. of MRI. Images |
|---|---|---|---|---|
| Meningioma | 82 | 708 | A * | 209 |
| | | | C * | 268 |
| | | | S * | 231 |
| Pituitary | 62 | 930 | A * | 291 |
| | | | C * | 319 |
| | | | S * | 320 |
| Glioma | 89 | 1426 | A * | 494 |
| | | | C * | 437 |
| | | | S * | 495 |
| Total | 233 | 3064 | | 3064 |

* Axial = A, * Coronal = C, and * Sagittal = S.

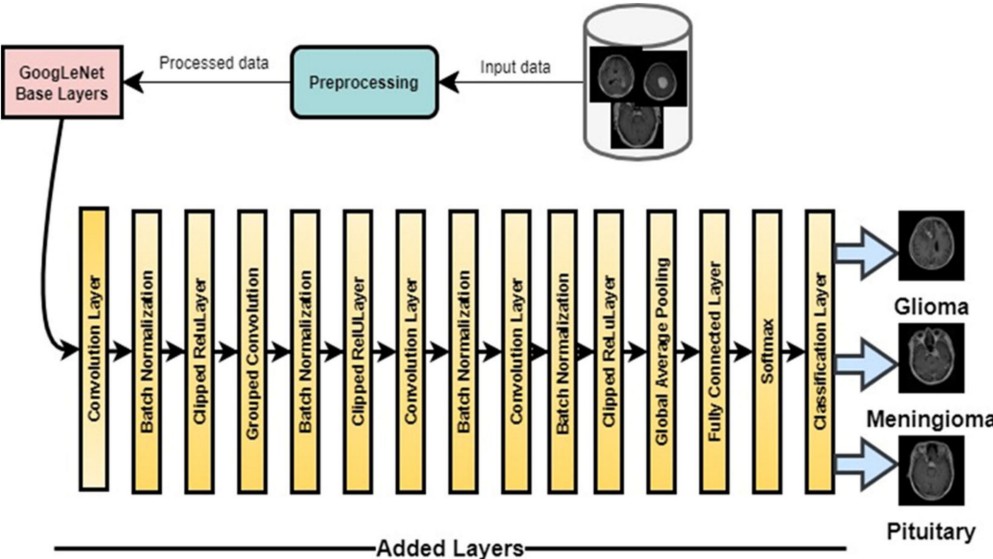

**Figure 2.** Architecture of proposed hybrid model.

### 3.2. DeepTumorNet

The hybrid DeepTumorNet model follows the basic structure of the CNNs. As training a convolution network from scratch is a challenging job, sometimes, it may take months [30,48]. Therefore, instead of training a new deep learning classifier from scratch, a pretrained classifier would be better to train the proposed deep learning approach. For this purpose, we used GoogLeNet as a base model, as GoogLeNet [26] remained the winner of the ILSVRC (2014) ImageNet competition. GoogLeNet has 22 learnable layers and 144 layers, including 2 convolution layers, 4 max_pooling layers, 1 average pooling layer, 2 normalization layers, 1 fully connected layer, and 9 inception layers modules [43]. Additionally, each inception module was comprised of one max-pooling and six convolutional layers. The input layer of GoogLeNet was renewed to 224 × 224 × 1. The ReLU activation function was used in the pretrained GoogLeNet approach. At the same time, the ReLU activation function neglected all negative values and used zero instead. Alternatively, Leaky ReLU is an improved version of ReLU, and it replaces all negative values with positive ones [58].

Meanwhile, in the proposed DeepTumorNet classifier, the last 5 layers of GoogLeNet were removed, and 15 new layers were added instead of 5 layers. Furthermore, the ReLU activation function in the feature map layer was changed to the Leaky ReLU activation function to increase the proposed model's expressiveness and overcome the dying ReLU problem without disturbing the primary convolution neural network architecture. The total count of layers after these changes increased from 144 to 154.

The first convolution layer used a filter (patch) size of $7 \times 7$, which immediately reduced the image size. The second convolution layer had a depth of two and leveraged the $1 \times 1$ convolution block, which is the effect of dimensionality reduction. Furthermore, the inception module of GoogLeNet has different convolution kernels, such as convolution kernels of $1 \times 1$, $3 \times 3$, and $5 \times 5$ which extract the features at different scales, starting from the most delicate features to the core features. The larger convolution kernel covers a larger area to compute the features. Similarly, the $1 \times 1$ convolution kernel gives further details and reduces the amount of computation. The new additions include four convolutional layers with a tiny filter size of $1 \times 1$. In addition, increasing the number of convolution layers in the CNN gave us more detailed, accurate, and robust features. These four later convolutional layers extracted high-level features compared with the initial layer, which extracted low-level features.

Moreover, the global average pooling layer increased the accuracy at the network's end. Furthermore, the ReLU activation function in the feature map layer was changed to the Leaky ReLU activation function to increase the proposed model's expressiveness and overcome the dying ReLU problem. Therefore, the proposed hybrid model extracted more detailed, discriminative, and deep features because of the additional layer, resulting in better classification performance than the mentioned state-of-the-art pretrained deep learning models.

The structure of the proposed hybrid DeepTumorNet approach is shown in Figure 2. The name, type, number of the filter, filter size, and epsilon of the added layer are elaborated in Table 2. Below, we discuss each layer of the proposed hybrid DeepTumorNet model in detail.

**Table 2.** Characteristics of added layers in the proposed hybrid model.

| S.No | Layer Name | Type | No of Filter | Filter Size | Epsilon |
|------|-----------|------|------------|-----------|---------|
| 1 | block_16_expand | Conv | 960 | $1 \times 1$ | |
| 2 | block_16_expand_BN | Batch Norm | | | 0.001 |
| 3 | block_16_expand_relu | Clipped ReLU Layer | | | |
| 4 | block_16_depthwise | Grouped Conv | 960 | $3 \times 3$ | |
| 5 | block_16_depthwise_BN | Batch Norm | | | 0.001 |
| 6 | block_16_depthwise_relu | Clipped ReLU Layer | | | 0.001 |
| 7 | block_16_project | Conv | 320 | $1 \times 1$ | |
| 8 | block_16_project_BN | Batch Norm | | | 0.001 |
| 9 | Conv_1 | Conv | 1280 | $1 \times 1$ | |
| 10 | Conv_1_bn | Batch Norm | | | 0.001 |
| 11 | out_relu | Clipped ReLU Layer | | | |
| 12 | global_average_pooling2d_1 | Global Average Pooling | | | |
| 13 | Logits | Fully Connected | | | |
| 14 | Logits_softmax | Softmax | | | |
| 15 | ClassificationLayer_Logits | Classification Layer | | | |

### 3.2.1. Image Input Layer

The proposed DeepTumorNet model starting from the image layer included the model's input, which specified the image input size, which in our case was $224 \times 224 \times 1$. Such a number corresponds to the input image's width, height, and channel size (1 in the case of grayscale images and 3 in the case of color images). The images were first read from in the input layer for processing.

### 3.2.2. Convolutional Layer

The convolutional layer was utilized to recover the deep learning features from an input image (producing a feature map). The mathematical operation presents two arguments: an image matrix and a filter size (size represents the height and width of the filters) [59]. Our hybrid model uses different filter sizes of $7 \times 7$, $5 \times 5$, and $1 \times 1$ in the convolutional layers and $3 \times 3$ in the max-pooling layers. Convolutional layers add padding with the

input of the feature map by using the "Padding" name–value pair. The discrete time convolution method is elaborated in Equation (1):

$$s(t) = (x * w)(t) = \sum_{a=-\infty}^{\infty} x(a)w(t-a) \tag{1}$$

where $W$ is the kernel filter, $x$ is the input to the method, $t$ is the time taken, and $s$ is the results. In the case of two-dimensional input data being taken, Equation (2) is considered:

$$S(i,j) = (I * K)(i,j) = \sum_{m} \sum_{n} I(i,j) * K(i-m,j-n) \tag{2}$$

The terms $i$ and $j$ show the areas of the desired matrix required after the deep learning convolution method. The preferred technique in this procedure is set so that the filter's center is in the first position.

If cross-entropy is to be accomplished in the proposed approach, Equation (3) is utilized:

$$S(i,j) = (I * K)(i,j) = \sum_{m} \sum_{n} I(i+m,j+n) * K(m,n) \tag{3}$$

### 3.2.3. Activation Function

The activation functions are often used in DL-based models for nonlinear transformation processes. Sigmoid, Tanh, and ReLU activation functions were the most widely used and preferred activation functions developed in the past. However, ReLU provides an output of zero for all negative inputs (i.e., deactivating negative inputs), resulting in dead neurons (the dying ReLU problem). A neuron is "dead" if it always outputs 0 and is stuck on the opposing side. We used the leaky ReLU activation function instead of ReLU in the feature map, an value-added form of the ReLU activation function, to address the dying ReLU problem [60]. In the case of leaky ReLU, the negative number (x) output is a tiny linear component of x instead of 0. Moreover, in the last 15 added layers, we used a clipped ReLU activation function which performed a thresholding operation, where any desired input value that was less than 0 was set to 0, and any desired input value that was above the ceiling was set to the specified ceiling. The formulas of the activation functions are shown in Equations (4)–(7):

ReLU:

$$f(x) = \begin{cases} 0, x < 0 \\ x, x \geq 0 \end{cases}, f(x)' = \begin{cases} 0, x < 0 \\ 1, x \geq 0 \end{cases} \tag{4}$$

Sigmoid:

$$f(x) = \frac{1}{1 + e^{-x}}, f'(x) = f(x)(1 - f(x)) \tag{5}$$

Tanh:

$$\tan h(x) = \frac{2}{1 + e^{-2x}} - 1, f'(x) = 1 f(x)^2 \tag{6}$$

Clipped ReLU:

$$f(x) = \begin{cases} 0, x < 0 \\ x 0 \leq x < ceiling \\ ceiling, x \geq ceiling \end{cases} \tag{7}$$

Leaky ReLU:

$$f(x) = \begin{cases} x, x \geq 0 \\ scale * x, x < 0 \end{cases} \tag{8}$$

In the leaky ReLU function, output x is for the positive inputs, and it outputs a small value that is 0.01 times $x$ in the case of negative values. Hence, no neuron is deactivated in this case, and we would no longer encounter dead neurons.

### 3.2.4. Batch Normalization Layer

The batch normalization layer was utilized to normalize the outputs generated by the proposed convolution layers. Normalization shortens the training time of the proposed DeepTumorNet model to achieve the learning process more efficiently and quickly. The batch normalization process is given in Equations (8)–(10):

$$Yi = \frac{Xi - \mu\beta}{\sqrt{\sigma^2\beta + \varepsilon}} \tag{9}$$

$$\sigma\beta = \frac{I}{M}(Xi - \mu\beta)^2 \tag{10}$$

$$\mu\beta = \frac{1}{M}\sum_{i=1}^{M} Xi \tag{11}$$

where $M$ is the total number of input data, $X\_i = 1, \dots, M$, $\mu\_\beta$ is the stack's average value, $\sigma\_\beta$ is the stack's standard deviation, and $Yi$ is the new values obtained as a result of the normalization procedure.

### 3.2.5. Pooling Layer

After the convolution layer, the pooling layer was used to simplify the information from the convolution layer (a downsampling procedure to decrease the size of the feature map and remove unnecessary data). Average and maximal pooling are the two most common pooling strategies. In the last 15 layers, we used global average pooling. In pooling, the network does not carry out any learning. For the pooling process, $3 \times 3$-sized filters were used. The pooling process is given in Equation (12):

$$S = w2 \times h2 \times d2 \tag{12}$$

$$w2 = \frac{(w1 - f)}{A + 1} \tag{13}$$

$$h2 = \frac{(h1 - f)}{A + 1} \tag{14}$$

$$d2 = d1 \tag{15}$$

where $w1$ represents the width of the images of the MRI, $h1$ denotes the height of the input MRI image, $d1$ denotes the value of the depth of the input MRI image size, $f$ represents the filter size, $A$ represents the number of steps utilized, and $S$ represents the size of the manufactured image.

### 3.2.6. Fully Connected Layer

In the proposed model, the convolutional layers are followed by a fully connected layer. This is accomplished by merging all of the features learned by the preceding layers over a number of images. This layer determines the most significant patterns in order to categorize the images. The output size value in the final completely linked layer is 3, as in the proposed research study, the number of classes (meningioma, glioma, and pituitary) is 3. For this, the obtained output value of the proposed FC layer is 3. Equations (16) and (17) are used for this purpose:

$$Ui^l = \sum_j wji^{l-1}yj^{l-1} \tag{16}$$

$$yi^l = f\left(ui^l\right) + b^{(l)} \tag{17}$$

where $l$ is the total number of the layers, $i$ and $j$ are the total number of neurons, $yli$ is the value created in the proposed output layer, $wl\text{-}1ji$ is the hidden layer's weight value, $yl\text{-}1i$ is the value of the input neurons, $uli$ is the output layer's value, and $b(l)$ is the value of deviation.

### 3.2.7. Softmax Layer

The activation function makes the output of the fully linked layer more normalized. Softmax executes the probabilistic calculation coming from the network and creates the work in positive numbers for each class. The Softmax method is given in Equation (18):

$$P(y = j|xi, W, b) = \frac{exp^{X^T Wj}}{\sum_{j=1}^{n} exp^{X^T Wj}} \qquad (18)$$

where *A*, *s*, *W*, and *b* are weight vectors.

### 3.2.8. Classification Layer

The last layer of the proposed model is the classification layer, which is used to produce the output by utilizing each input. The Softmax activation function returned a probability distribution [61].

### 3.2.9. Training Parameters

We used a trial-and-error-based approach to perform experiments with the parameters shown in Table 3. We constantly monitored the improvement of the training validation accuracy and error to find the optimal convergence of each CNN. If there was no validation accuracy or error increase, the training was terminated automatically. We used stochastic gradient descent (SGD) to train the proposed DeepTumorNet model with an initial and final learning rate of 0.01 and a minibatch size of 10 images. The proposed DeepTumorNet model was trained on 120 epochs for brain tumor classification to obtain the optimum results.

**Table 3.** Parameter values used in training networks.

| Name | SGDM |
|---|---|
| MiniBtachSize | 10 |
| Number of Epochs | 120 |
| Initial Learning Rate | 0.01 |
| Shuffle | every epoch |
| Validation Frequency | 50 |

## 4. Results and Discussion

This research paper was intended to classify three different types of BTMRI images correctly. The CEMRI dataset was classified with pre-trained deep learning models and the proposed hybrid DeepTumorNet model, where 70% of the dataset was used for training purposes and 30% for testing purposes. The results were achieved in the Matlab environment with computer resources, namely an i5 processor and 8 GB of RAM. There are various approaches for the measurement of deep learning network classification performance. A confusion matrix is widely used for classification tasks in the CNN process. The most desired and preferred measures included precision, recall, accuracy, and F1 score. These calculations are computed through a confusion matrix [62]. The general form of the confusion matrix utilized in this research is shown in Table 4.

**Table 4.** General confusion matrix.

| | | Predicated Classes | | |
|---|---|---|---|---|
| | | Gliomas | Meningiomas | Pituitary |
| **Actual Class** | Gliomas | PGG | PMG | PPG |
| | Meningiomas | PGM | PMM | PPM |
| | Pituitary | PGP | PMP | PPP |

Here, PGG is the images from the dataset which predicted glioma; PMG is the images from the dataset which were actually meningioma and predicted wrongly as glioma; PPG is the images from the dataset which were actually pituitary and predicted wrongly as glioma; PGM is the images from the dataset which were actually glioma and predicted wrongly as meningioma; PMM is the images from the dataset which were actually meningioma and predicted correctly as meningioma; PPM is the images from the dataset which were actually pituitary and predicted wrongly as meningioma; PGP is the images from the dataset which were actually glioma and predicted wrongly as pituitary; PMP is the images from the dataset which were actually meningioma and predicted wrongly as pituitary; and PPP is the images from the dataset which were actually pituitary and predicted correctly as pituitary. Furthermore, the accuracy, precision, recall, and F1-score are defined below.

### 4.1. Performance Metrics (Accuracy, Precision, Recall, and F1 Score)

Classification accuracy is the proportion between the correct predictions and the total data elements.

The equation that calculates the accuracy is shown in Equation (19) and Figure 3:

$$Acc = \frac{TP + TN}{TP + FP + TN + FN} \tag{19}$$

where *TP* is true positive, or the estimated accurate amount of data, *FP* is false positive, or the fact that it is harmful and predicated as positive, *TN* is true negative, which is genuinely harmful and is predicated negatively, and *FN* is false negative, or the information that it is positive and predicated as unfavorable.

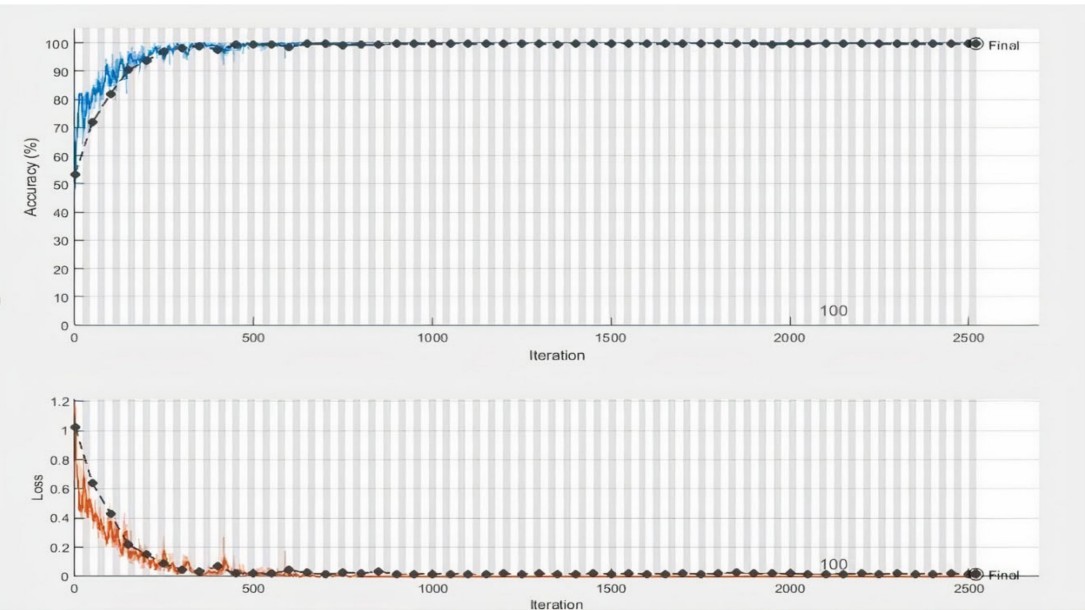

**Figure 3.** Classification performance (accuracy (%)) of the proposed hybrid model.

Precision (positive predicted values) is the proportion of optimistic forecasts that belong to the all positive class. The equation of the precision value is shown in Equation (20):

$$prec = \frac{TP}{TP + FP} \tag{20}$$

Recall is defined as the number of true positive (*TP*) outcomes divided by the total number of elements corresponding to the positive class. The calculation for the recall value is presented in Equation (21):

$$Recall = \frac{TP}{TP + FN} \tag{21}$$

The F1 score is the average of the precision and recall. The calculation for the F1 score is presented in Equation (22):

$$F1 - measure = \frac{2 * prec * recall}{prec * recall} \tag{22}$$

Additionally, after the proposed hybrid DeepTumorNet model was trained, the performance values of the accuracy, precion, recall, and F1 score were measured as presented in Table 5. The proposed hybrid deeptumorNet model obtained the highest accuracy, precision, recall, and F1 score values of 99.67%, 99.6%, 100%, and 99.6%, respectively. The loss and accuracy curves of the proposed DeepTumorNet approach are shown in Figure 3. The loss function shows how well the proposed hybrid model classified the tumor images into fine-grained tumors (gliomas, meningioma, and pituitary). The loss and accuracy of the proposed model after epoch 71 almost remained the same, demonstrating that the proposed hybrid approach classified the brain tumors with higher accuracy even at lower epochs than 120. The training and validation process of the proposed DeepTumorNet model is presented in Figure 3. A confusion matrix is shown in Table 6, which summarizes the correct and incorrect classifications of our proposed method.

**Table 5.** Experimentation results of pretrained models.

| Model | Accuracy | Precision | Recall | F1-Score |
|---|---|---|---|---|
| Proposed Model | 99.67% | 99.6% | 100% | 99.66% |
| AlexNet | 97.8% | 97.6% | 97.66% | 97.66% |
| GoogLeNet | 98.26% | 98% | 98.66% | 98.33% |
| Shufflenet | 98.37% | 98.33% | 98.66% | 98.33% |
| ResNet50 | 98.60% | 98.33% | 98.66% | 98.33% |
| MobileNet V2 | 99% | 99% | 99% | 99% |
| SqueezeNet | 97.91% | 97.66% | 98% | 97.66% |
| Darknet53 | 99.13% | 99% | 99.33% | 99% |
| Resnet101 | 98.91% | 98.66% | 99% | 98.66% |
| ExceptionNet | 98.69% | 98.33% | 98.33% | 98% |

**Table 6.** Confusion matrix of proposed hybrid model.

| | | Predicated Classes | | |
|---|---|---|---|---|
| | | **Gliomas** | **Meningiomas** | **Pituitary** |
| **Actual Class** | Gliomas | 426 | 01 | 01 |
| | Meningiomas | 01 | 211 | 00 |
| | Pituitary | 00 | 00 | 278 |

### 4.2. Comparison of DeepTumorNet with the Pretrained Transfer Learning Approaches

This section compares some pre-trained transfer learning models with the proposed hybrid DeepTumorNet model. This experimentation aimed to evaluate the usefulness of the hybrid proposed model for BTs classification. At the same time, we compared the evaluation performance of our DeepTumorNet model with nine classical DL models (i.e., ExceptionNet, MobileNetv2, SqueezeNet, ShuffleNet, DenseNet, ShuffleNet, ResNet50, MobileNetv2, DarkNet-53, ResNet101, and AlexNet). All of these comparable DL models were employed using a transfer learning configuration trained on the ImageNet database. We changed the last three layers of each pre-trained model to adapt them to the target

number of classes. The fully connected (FC) layer in every model was removed, and a new FC layer having an output size of three was introduced using the fine-tuning strategy, as we had three classes as output. Every model had a different input size, and like the size of the input images, GoogLeNet's was 224 × 224, while 227 × 227 was the for SqueezeNet and other models. All models were fine-tuned with the same experimental parameters shown in Table 6. The fine-tuning classification method is illustrated in Figure 4. To perform the deep learning training, the dataset's images were split at a ratio of 30% and 70% for testing and training, respectively, to obtain reliable results and efficient performance for the deep neural networks as shown in Figure 3. The results are presented in Table 5.

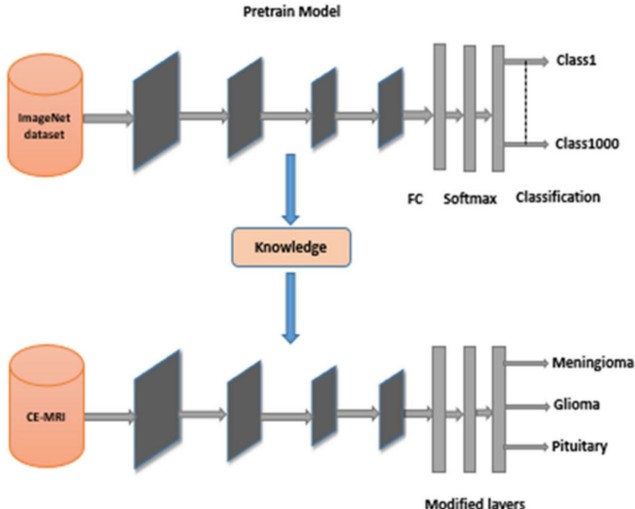

**Figure 4.** Diagram of transfer learning-based classification.

We witnessed that our method expressively outperformed the nine comparative DL models from the resulting observations. However, the Darknet53 and MobileNetV2 pretrained classifiers performed relatively better and close to the proposed hybrid DeepTumorNet model by achieving the second and third highest performance in classifying brain MRI images to meningioma, glioma, and pituitary types.

*4.3. Comparative Results of DeepTumorNet with the State-of-the-Art Classification Approaches*

Previously, researchers conducted relevant studies on BTs using different machine learning and DL techniques with a similar dataset. However, in some studies, the authors used binary classification classes (malignant and benign) of data and multiple classes (fine-grained classification) with fewer images. It is noticeable in Table 7 that the proposed deep learning-based hybrid model's performance values were enhanced compared with the state-of-the-art approaches in the literature.

We compared the obtained results of our proposed DeepTumorNet approach with the state-of-the-art approaches in the literature [27,30,31,35,56,57]. The comparative results indicated the greater efficiency of the hybrid DeepTumorNet method over these techniques. Additionally, it was essential to use hand-crafted engineering for classification, which is computationally more complex. Moreover, the proposed model performed well than the mentioned methods in terms of accuracy, as depicted in Table 7. The proposed hybrid model comprised 154 layers, including newly added layers followed by the leaky ReLU activation function, decreasing the dying ReLU problem. Therefore, the proposed hybrid model extracted more descriptive and discriminative details and accurate features for classification.

In the proposed hybrid model, GoogLeNet was modified by adding and replacing the last 5 layers with 15 new layers and replacing the ReLU activation function with leaky ReLU. The newly proposed method performed better than the base model GoogLeNet in classification accuracy. The DeepTumorNet model is a convolutional neural network architecture with fewer hardware resources. It also requires a very convenient time for

training. The increasing number of epoch and dataset sizes increased the time complexity of the network. However, the proposed deep learning-based hybrid model extracted more descriptive, detailed, and discriminative deep features. The proposed model can be embedded in an MRI machine for real-time BT classification. Moreover, it will help neurologists and surgeons in the treatment of BT patients.

**Table 7.** Comparative study of the proposed model with recent ML and DL models.

| Author | Technique | Classification Type | Dataset | Accuracy (%) |
|---|---|---|---|---|
| Mehrotra et al., (2020) [27] | PT-CNN: AlexNet | Binary Class | T1-weighed MRI (Benign = 224; malignant = 472) | 99.04% |
| Kaplan et al., (2020) [35] | LBP SVM KNN | Multi-Class | T1-weighed CE-MRI (meningiomas = 708; gliomas = 1426; pituitary = 930) | 95.56% |
| Sultan et al., (2019) [30] | CNN | Multi-Class | T1-weighed CE-MRI (meningiomas = 208; gliomas = 492; pituitary = 289) | 96% |
| Anaraki et al., (2019) [63] | GA-CNN | Multi-Class | T1-weighed CE-MRI (meningiomas = 708; gliomas = 1426; pituitary = 930) | 94.20% |
| Kumar et al., (2017) [31] | GWO+M-SVM | Multi-Class | T1-weighed CE-MRI (meningiomas = 248; gliomas = 12; pituitary = 55) | 95.23% |
| Bahadur et al., (2017) [64] | BWT+SVM | Binary | T2-weighted images (normal = 67; abnormal = 134) | 95% |
| Abiwinanda et al., (2019) | CNN | Multi-Class | T1-weighed CE-MRI (meningiomas = 708, gliomas = 1426; pituitary = 930) | 84% |
| Proposed method | Deep CNN | Multi-Class | T1-weighed CE-MRI (meningiomas = 708; gliomas = 1426; pituitary = 930) | 99.67% |

## 5. Conclusions and Future Work

This research intended to classify BTs using different convolution neural networks and a new hybrid model. The GoogLeNet architecture was utilized as a base for the proposed DeepTumorNet framework. The last 5 layers of GoogLeNet were eliminated and 15 new deep layers were added in place of the 5 layers. Furthermore, the ReLU activation function was changed into the leaky ReLu activation function without disturbing the primary convolution neural network architecture. The total count of layers after the changes increased from 144 to 154. The proposed hybrid model reached a highest ever classification accuracy of 99.67%. In addition, we deployed nine deep pretrained CNN models using the transfer learning technique on the CE-MRI dataset to identify the BT types and compared their results with the proposed hybrid model. The experimental results demonstrated that the proposed hybrid model more accurately discriminated the brain tumors. Moreover, the proposed method computed more descriptive and discriminative details and accurate features for brain classification, resulting in high accuracy compared with the other state-of-art approaches.

Furthermore, it is evident from experimentation that the pretrained CNN model using transfer learning techniques produced the utmost performance. However, the hybrid framework accuracy reached the maximal compared with the rest of the pretrained models. Furthermore, in future work, experimenting with the dataset with a small number of malignant brain MRI images and a significant number of normal brain MRIs should be performed, as the proposed model extracted more detailed, discriminative, and accurate features. Therefore, before classifying the brain MRI images into two classes (i.e., malignant and benign), an efficient segmentation technique should be applied to brain MRI data. After that, the proposed model can accurately detect and classify benign and malignant images using segmented images.

Additionally, there is a possibility that other CNN networks can be converted to hybrid approaches to show better classification results with less time complexity. We aim to classify

BTCT images and other large BT datasets [65] with the proposed DeepTumorNet model. Moreover, we plan to check the efficiency of the proposed hybrid method for other forms of medical image analysis, such as lung cancer, COVID-19, and pneumonia detection.

**Author Contributions:** Conceptualization, A.R.; methodology, A.R.; software, A.R., validation, A.R., H.A. and I.A.; formal analysis, A.R., D.J., J.A.K., I.A., Y.I.D. and A.S.S.; investigation, I.A. and A.U.R.; resources, H.A., J.A.K., Y.I.D., A.U.R. and H.H.; data curation, A.S.S., I.A., D.J., Y.I.D. and A.U.R.; writing—original draft preparation, A.R. and I.A.; writing—review and editing, I.A., Y.I.D., A.U.R. and H.H.; visualization, J.A.K. and A.U.R.; supervision, I.A. and J.A.K.; project administration, I.A., Y.I.D. and H.H.; funding acquisition, I.A., A.S.S. and H.H. All authors have read and agreed to the published version of the manuscript.

**Funding:** This research was supported by Future University Researchers Supporting Project Number FUESP-2020/48 at Future University in Egypt, New Cairo 11845, Egypt.

**Data Availability Statement:** The datasets used in this investigation are available on request from the corresponding author.

**Conflicts of Interest:** The authors declare no conflict of interest.

## Abbreviations

The following abbreviations are used in this paper:

| | |
|---|---|
| AI. | Artificial intelligence |
| ML | Machine learning |
| DL | Deep learning |
| TL | Transfer Learning |
| BT | Brain tumors |
| MRI. | Magnetic resonance imaging |
| K-NN | K-nearest neighbor |
| SVM | Support vector machine |
| CNN | Convolutional neural network |
| PCA. | Principal component analysis |
| LDA | Linear discriminant analysis |

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
