# Peer review of "A Hybrid Deep Learning-Based Approach for Brain Tumor Classification"

_electronics, doi:10.3390/electronics11071146_

Round 1

Reviewer 1 Report

The manuscript entitled” A Hybrid Deep Learning-based Approach for Brain Tumor Classification“ has been investigated in detail. The topic addressed in the manuscript is really very interesting and applicable and the manuscript contains some practical meanings. The presentation of the manuscript is very well and the literature of the manuscript is good however it could be improved. There are a few issues which could be addressed by the authors:

  • Authors state that: the last five layers of GoogLeNet have been removed, and 15 new layers have been added instead of these five layers. The authors should be described clearly what is the reason of this suggestion and how they optimized the mentioned neural network.

  • Figure 3 is unclear and needs more clarification. It should be modified.

  • “Roshani, M. et al. Evaluation of flow pattern recognition and void fraction measurement in two phase flow independent of oil pipeline’s scale layer thickness. Alex. Eng. J. 2021, 60, 1955–1966, doi:10.1016/j.aej.2020.11.043.” and “Sattari, M.A.et al. Applicability of time-domain feature extraction methods and artificial intelligence in two-phase flow meters based on gamma-ray absorption technique. Measurement 2021, 168, 108474, doi:10.1016/j.measurement.2020.108474.” could be used in the study.

  • How the best number of epochs was obtained? Please add to the text.

This study may be consider for publication.

Author Response

Many thanks to you and your reviewer team for reviewing our manuscript (Electronics-1657253) entitled A Hybrid Deep Learning-based Approach for Brain Tumor Classification.” The reviewers’ comments were beneficial to improve the quality of our manuscript, and therefore we revised our manuscript accordingly. As suggested by the reviewers, we modified the original manuscript and edited it thoroughly.

Reviewer 2 Report

The authors provide a fascinating deep learning-based technique for brain tumor categorization. I highly recommend that this paper be published in the journal Electronics. However, I have a few concern about result, and the writers may need to alter their claim in light of the following considerations.

1) Binary classification using a neural network is used to distinguish between normal and malignant cases. In the normal case, all tiles are deemed normal, but all tiles in the malignant condition are considered tumors. When considering that a tumor sample may have a big number of normal tiles but only a small number of tumor tiles, how can this method be justified?

2) For hyperparameter adjustment and optimum model selection, deep learning-based techniques often need a validation set in addition to a training set. It is unclear, however, how they chose the best models for the subsequent this studies over Alex net, Resnet50, darknet53, Shufflenet, GoogLeNet, SqueezeNet, ResNet101, Exception Net, and MobileNetv2.

3) DNN models have emerged as strong tools in a variety of sectors where older algorithms have failed to attain acceptable competency. The convolutional CNN model is the most extensively used form of DNN model, with results in applications as diverse as voice recognition, reinforcement learning, and text translation. Authors may need to mention such studies in the introductory section of this publication using references ( Appl. Sci. 2021, 11(8), 3603; Small 2021, 17, 2103543; Sci Rep 9, 874 (2019)

Author Response

(The authors gave the same response as above.)
